# Loneliness and Psychosocial Resources among Indigenous and Afro-Descendant Older People in Rural Areas of Chile

**DOI:** 10.3390/ijerph20032138

**Published:** 2023-01-24

**Authors:** Lorena P. Gallardo-Peralta, José Luis Gálvez-Nieto, Paula Fernández-Dávila, Constanza Veloso-Besio

**Affiliations:** 1Department of Social Work, Universidad Alberto Hurtado, Santiago 8320000, Chile; 2Department of Social Work, Universidad de La Frontera, Temuco 4811230, Chile; 3Department of Social Work and Social Services, Universidad Complutense de Madrid, 28040 Madrid, Spain; 4School of Psychology, Universidad de Tarapacá, Arica 1000007, Chile

**Keywords:** indigenous, Afro-descendant, older adults, loneliness, family functioning, resilience, main health problems

## Abstract

(1) Background: loneliness is a problem that becomes increasingly acute in old age, with greater repercussions among socially disadvantaged groups such as indigenous and Afro-descendant older adults. The aim of this research is to analyze the psychosocial variables related to loneliness in old age. (2) Methods: a multi-ethnic sample was involved, with the participation of eight indigenous peoples and Afro-descendant tribal people (n = 1.348). Various gerontological scales previously validated among the Chilean population (De Jong Gierveld Loneliness Scale, Brief Resilient Coping Scale, Health Problems Questionnaire, and Family APGAR questionnaire) and a model are contrasted, establishing the relationship between psychosocial variables and loneliness. (3) Results: Structural equation modeling (SEM) showed the existence of indirect relationships between health problems, via family functioning and resilience, and loneliness. Resilience and family functioning were directly related to loneliness (WLSMV-χ^2^ (*df* = 345) = 875.106, *p* < 0.001; CFI = 0.992; TLI = 0.991; RMSEA = 0.034 [C.I. 90% = 0.031–0.037]). (4) Conclusions: loneliness has cross-culturally affected older Chilean people living in rural areas and it appears that the COVID-19 pandemic has had a negative effect on well-being. This study proves that loneliness is related to several psychosocial variables that can be intervened.

## 1. Introduction

### 1.1. Loneliness among Older Adults

Loneliness is defined as a public health problem [1,2,3] given its high prevalence and relationship to quality of life, meaning that it requires monitoring [2]. One recent estimate suggests that one-third of the population in industrialized countries experiences loneliness, with one in twelve people reporting problematic levels [4].

Loneliness can be defined as a discrepancy between desired and existing qualities or quantities of social relationships [5]. It is therefore a subjective and negative experience resulting from deficits in social networks. People experience dissatisfaction with their social relationships because they have fewer links than they desire or because their existing links do not offer the expected intimacy, affection, and value [6,7]. Newmyer et al. [8] affirm the characteristics of loneliness can be summarized as follows: (a) it is the outcome of deficits in social relationships; (b) it is subjective; and (c) it is unpleasant and stressful. Everyone experiences transient loneliness [9], but chronic or serious loneliness threatens health and well-being [2].

Weiss [10] states that loneliness can be prevented by having a network of social relationships that encourages social integration, self-worth, and a sense of direction. The existence of loneliness is hence more closely linked to the emotional quality of contact than to the amount of contact per se [11]. Following this line of argument, Weiss [10] distinguishes between “social loneliness” and “emotional loneliness”, where the former refers to the lack of a network of social relationships, and the latter concerns the lack of emotional intimacy in close relationships [12]. The experience of loneliness is thus established as a phenomenon that is different from objective social isolation, defined as an absence of social interactions and a personal tendency to be alone [13]. There is no causal relationship between social isolation and the experience of loneliness: people may be part of an extensive social network but experience feelings of loneliness, while at the same time, there can be solitary people who do not experience loneliness [1,13].

Loneliness is a problem that can be experienced at any stage of life [1,3]. However, older adults are a more vulnerable group in this regard owing to various factors [14,15] including declining social networks, the presence of medical issues, limitations affecting functionality, changes in position or social status, and the loss of relatives and/or friends [11]. A scientific consensus has developed over the last two years regarding the negative effects of the COVID-19 pandemic in terms of increased loneliness and social isolation among older adults [1,3,16].

### 1.2. Risk Factors in Terms of Triggering Loneliness in Old Age

Loneliness is a social determinant of health [17], making it important to examine its contributory factors as there are causes of loneliness that might be preventable or changeable [18]. In this regard, Victor et al. [19] identified six factors of vulnerability among older adults: marital status, increases in loneliness over the previous decade, increases in time alone over the previous decade, elevated mental morbidity, poor current health, and poorer-than-expected health in old age. These authors hence recognized the importance of examining life course to understand loneliness in old age. The life course approach makes it possible to examine the role of early, mid, and later-life events to understand the patterns of loneliness in later life.

A study conducted by Kamiya et al. [18] adopted a life course perspective to analyze how circumstances in early life (family situation and financial well-being, urban/rural residence, childhood health, and parental substance abuse) can have a direct impact on loneliness in later life or predispose individuals to experience financial, health, and social circumstances in later life that then contribute to loneliness. This study found that poor childhood socioeconomic status (for men and women) and parental substance abuse (for men) had direct effects on loneliness at older ages. 

The problem of loneliness is more acute among more vulnerable groups. The 2016 social report for Aotearoa/New Zealand, which is described as a state-of-the-nation report for social well-being [20], reported that women, Maoris, and Asian respondents declared higher levels of loneliness. Similar findings are observed in studies conducted in other countries, with high incidences of loneliness among women [21] and groups comprising ethnic minorities [22,23].

Various studies confirm that indigenous ethnic minorities have more difficult life experiences that are determined by inequality gaps in terms of health, education, working conditions, racism, forced migration, and confiscation of and estrangement from ancestral lands, among other high-risk experiences [24,25,26,27]. This makes it unsurprising that indigenous older adults experience old age with greater social disadvantages and can experience higher levels of loneliness or a sense of loss of social integration [26,28,29,30]. However, one of the strongest determining factors of loneliness among indigenous older adults is a change in household structure [26,29,30], taking into account that the system organized around extended family and intergenerational care is changing and there are increasing numbers of single-member households or older couples living alone. Increased levels of institutionalization have also been observed among indigenous older adults. In this respect, a systematic review conducted by Chen et al. [31] reported that indigenous older adults living in residential care facilities experienced more loneliness.

There have been few studies in Latin America examining the problem of loneliness among indigenous older adults; although, concerns have been raised that changes in close social relationships are affecting the well-being of this social group. In this regard, Waters and Gallegos [32] observed indigenous communities in Ecuador experiencing changes in household structure involving a trend toward smaller families, with indigenous older adults reporting a general perception of weaker family networks. 

Unexpectedly, the physical environment can act as a buffer against loneliness—that is, the opportunity to age in one’s native territory, and thereby maintain traditional forms of economic, social, and family organization. Along these lines, a study by Sánchez et al. [28] compared the prevalence of loneliness among Chilean older adults living in rural and native indigenous settings, observing higher levels of loneliness among non-indigenous people (average 4.05), followed by indigenous Mapuche participants (average 3.07) and indigenous Aymara participants (average 2.70). 

In Chile, the study of loneliness in old age is relatively recent and the data show risk factors in this age group. Regarding the non-indigenous and ageing population, the study conducted by Carrasco et al. [33] in the city of Santiago, using the UCLA abbreviated scale, found 45% to perceive feelings of loneliness at least some of the time. This study showed an association between loneliness and family dysfunction, depressive symptoms, living alone, not having a partner (widowed, separated, or single), having little contact with relatives and friends, feeling a lack of social support, and sensation of poor self-efficacy.

### 1.3. Present Piece of Research 

Currently, 9.5% of Chile’s population identify as descendants of indigenous peoples. In this respect, it is interesting to note the recent increase in indigenous self-identification: the corresponding percentage in 2006 was 6.6%, and the percentage of 9.5% was reached in 2017. Law 19,253 provides a legal basis for the recognition of ten indigenous peoples: Aymara, Atacameño, Colla, Quechua, Rapa Nui, Mapuche, Yámana, Kawashkar, Diaguita, and Chango. Recognition has been granted to Afro-descendant tribal peoples since 2019 (Law 21,151) [34]. Given the recency of their political and legal recognition, there are no official data for this ethnic group and they will be included in the next national census that is due in 2024.

This study analyzed loneliness among indigenous and Afro-descendant older adults living from the far north (the border with Peru and Bolivia) to the far south (Magallanes and Chilean Antarctica), including Rapa Nui. The minimum age for participants in the study was set at 60 years, since in Latin America in general, and in Chile in particular, this is the age at which institutions generally set the age at which a person is considered to be an older adult. Few studies have addressed the prevalence and variables associated with loneliness during ageing in Chile, and these studies are of non-indigenous people living in urban areas [33,35]. Furthermore, there is evidence of how loneliness manifests itself in only two of the eleven recognized indigenous or Afro-descendant tribal peoples [28,36].

The aim of this study is to analyze the psychosocial variables that are related to loneliness among older adults. Based on the premise that indigenous and Afro-descendant older adults are highly resilient and place greater emphasis on their family ties, it is proposed that resilience and family functioning mediate the relationship between physical health problems, specifically diagnosed chronic illnesses, and loneliness. It is also hypothesized that family functioning and resilience have a direct and negative relationship with loneliness. These hypotheses are represented in Figure 1. 

## 2. Materials and Methods

### 2.1. Participants

The study was based on a national and cross-sectional study entitled: “*Ethnic diversity and ageing: producing a multicultural map of successful ageing in Chile*”. The sample consisted of 1348 older people living in nine regions of Chile, ranging from the north to the south of the country: Arica and Parinacota, Tarapacá, Antofagasta, Atacama, Coquimbo, Valparaíso, Los Lagos, Aisén, and Magallanes. The inclusion criteria were age (60 years or older), residence (rural setting), health status (no severe cognitive impairment), and voluntary participation in the study. A sample stratified by sex, ethnicity, and place of residence (municipal or rural areas) was used to ensure representativeness in each of the aforementioned territories, and an ethnic quota was established for each study region (which was calculated by the 2017 Census data). This is a non-probabilistic and convenience sample, since, as will be indicated below, the sample collection process was affected by various problems that affected the randomness and, nevertheless, the quotas established for each ethnic group were reached.

The general characteristics of the sample are presented in Table 1. Notably: 57% were women, with 49% aged between 60 and 69 years (70.67 ± 7.44), 20% living alone, 53% were married or had a (cohabiting) partner, and 48% had completed high school and/or vocational education. The multi-ethnic sample was distributed as follows: 203 Quechua, 214 Atacameño, 215 Colla, 100 Chango, 255 Diaguita, 130 Rapa Nui, 116 Huilliche, 10 Kawésqar, and 105 Afro-descendants.

### 2.2. Procedure

A pilot study was applied with 45 individuals, comprising five from each of the aforementioned peoples. The questionnaire was applied in Spanish with certain indigenous-language terms. This phase of the study was more rigorous because preliminary work was carried out with experts (anthropologists, sociologists, and social workers) in each case, as well as with indigenous leaders, particularly to develop certain sections of the questionnaire.

A face-to-face interview method was used to collect the data with the questionnaire read out loud to interviewees. Qualified social work and psychology professionals administered the questionnaire after a short training workshop at which they received instructions on how to address potential difficulties in understanding questions. The professionals met with respect, trust, and openness among older adults and this enabled them to conduct most interviews at participants’ homes. It was only necessary to establish a meeting point on a few occasions, with the rural local council usually the chosen venue (an office offering interviewees the necessary confidentiality and peace of mind).

The study was applied over a longer than customary period of time (from December 2021 to June 2022) owing to social, socio-healthcare, and demographic difficulties. With respect to the former, there was a change in levels of citizen trust following the “social outburst” of 18 October 2019 in Chile, which involved a series of protests at the accumulation of social inequalities in the country in the form of increased prices for basic services and ballooning debt for working and middle-class families. These protests saw numerous deaths and injuries, particularly to eyesight, in addition to reports of police abuse. The outcome was the social/political agreement to amend the 1980 Chilean Constitution. This background of social protest has given rise to a change in levels of trust among Chilean citizens, including older adults, who displayed a more cautious and mistrusting attitude when participating in this study. Participants demanded greater guarantees of their rights in the research (particularly the right to anonymity), which made it necessary to more carefully safeguard the ethical aspects of the research procedures. The socio-healthcare factors were related to the fact that the study was conducted during the COVID -19 pandemic, which entailed observing all of the security measures required by the Ministry of Health of the Chilean government, including completion of the vaccination program for interviewers, social distancing, use of masks, and use of alcohol gel for tasks including the signing of the consent form. The fieldwork on Rapa Nui was altered because the island was closed (with no commercial flights) since the indigenous community had declared *tapu*, the concept of self-preservation and safeguarding of the community against adversity. This required respect for the decisions made by the traditional authorities and the territory hence remained closed to third parties [37]. As a result, interviews were conducted with older adults who had traveled to a healthcare facility in the city of Valparaíso, to be attended to for non-COVID-19 health issues in a more advanced hospital. Finally, demographic factors changed the balance of the sample in terms of peoples from the far south of Chile (regions close to the Antarctic), living in the so-called fjords and channels of Chile. Both the Yaghan and Kawésqar peoples have experienced significant drops in population to the extent that they are on the verge of extinction, and this meant that only ten older adults were interviewed.

### 2.3. Ethical Issues

The Ethics Committee of Tarapacá University supervised and approved the ethical aspects of both studies (monitoring reports nos. 21/2021 and 01/2022). All procedures in studies involving human participants were conducted in accordance with the Declaration of Helsinki 1964 and the regulations established in the Indigenous and Tribal Peoples Convention of the International Labor Organization (Convention 169). Not every indigenous community in Chile has yet granted express legal permission for research to be conducted. However, various indigenous community leaders were invited to take part in this study throughout the research process.

### 2.4. Measures

Loneliness. The 6-item version of the De Jong Gierveld Loneliness Scale (DJGLS-6) was used [7]. Items are scored on a scale from 0 to 2 and then recoded as dichotomous (0 or 1). The final score ranges from 0 (no loneliness) to 6 (extreme loneliness), with two categories: no loneliness (scores: 0–1) and loneliness (score equal to or greater than 2). The Spanish version has been previously validated among Chilean older adults using a multi-ethnic sample, including indigenous older adults [36], with an internal consistency index (Cronbach’s alpha) of 0.73.

Resilience. The Brief Resilient Coping Scale (BRCS) [38], is composed of four items ranging from 1 (does not describe me at all) to 5 (describes me very well). The total score ranges between 4 and 20. A total score lower than or equal to 13 indicates low resilience, while scores equal to or greater than 17 indicate high resilience. This questionnaire has been validated among the Chilean population [39], with an internal consistency index (Cronbach’s alpha) of 0.91.

Main health problems. These problems were assessed using the Health Problems Questionnaire produced by Herrera et al. [40]. This instrument was specifically developed to measure the most recurrent illnesses among the older adult population in Chile, within the framework of the Chilean National Survey of Quality of Life in Old Age. The instrument comprises an inventory/checklist made up of 13 pathologies: tension or hypertension; arthritis; high cholesterol; diabetes or elevated blood sugar levels; cataracts; osteoporosis; heart problems; chronic lung disorders; stomach ulcers; asthma; fractured hip or femur; cancer; stroke or vascular disorders; and Parkinson’s disease. 

Family functioning. The Family APGAR questionnaire [41] is a standardized questionnaire used to measure family functioning. It consists of five items that evaluate an individual’s perception of the support offered by their family: adaptation, partnership, growth, affection, and resolve. The final score of the questionnaire ranges from 0 to 10 and the cut-off points are as follows: severely dysfunctional family (0–3 points); family with mild dysfunction (4–6 points); and functional family (7–10). This questionnaire has been validated among the Chilean population [42], the internal consistency index (Cronbach’s alpha) for the general questionnaire was 0.90.

### 2.5. Analysis

Centralization and dispersion measures were estimated and univariant and multivariant normality tests were carried out to fulfill the study aim. Confirmatory factor analysis (CFA) and structural equation modeling (SEM) were subsequently performed using the MPLUS v.8.1 software [43]. Specific estimation methods were implemented for categorical variables: polychoric correlations matrix and the weighted least squares means and variance adjusted (WLSMV) estimation method. The following goodness-of-fit indexes were used for the evaluation of the statistical models: WLSMV- χ2, comparative fit index (CFI), Tucker–Lewis index (TLI), and root mean square error of approximation (RMSEA). For the CFI and TLI, values equal to or greater than 0.90 were considered adequate [44]. Values lower than or equal to 0.080 were considered a reasonable fit for RMSEA [45].

## 3. Results

### 3.1. Descriptive and Correlation Analyses

Table 2 shows the frequencies, broken down by ethnic group, from dichotomizing the loneliness scale (DJGLS-6) into no loneliness and loneliness categories. A high general prevalence of loneliness was observed among indigenous and Afro-descendant older adults (≥55%), with the exception of low prevalence among Rapa Nui older adults (9%) and Diaguita (14%).

Table 3 shows the descriptive results for the latent variables. The highest median was observed for the family functioning variable (median = 0.596), and the lowest was observed for the health problems variable (median = −0.284). It was also observed that none of the latent variables fit the K-S test normal distribution *p* < 0.001.

Table 4 shows the results of the correlations between the factors. In general, statistically significant moderate correlations were observed between all the variables, with a strikingly low correlation coefficient between health problems and family functioning (r = −0.065; *p* < 0.001).

### 3.2. Complete Structural Model

Figure 2 presents the results of the complete structural model, which included relationships between all the latent variables. The results of this model presented satisfactory values (WLSMV-χ^2^ (*df* = 269) = 829.474, *p* < 0.001; CFI = 0.991; TLI = 0.990; RMSEA = 0.036 [C.I 90% = 0.038–0.042]). As may be observed from Figure 2, the model presented statistically significant negative relationships between the variables of health problems and resilience (β = −0.529; *p* < 0.001), resilience and loneliness (β = −0.250; *p* < 0.001), and family functioning and loneliness (β = −0.580; *p* < 0.001), and a statistically significant positive relationship between resilience and family functioning (β = 0.459; *p* < 0.001). No statistically significant relationship was found between health problems and loneliness (β = 0.105; *p* = 0.100) and health problems and family functioning (β = −0.100; *p* = 0.068). To improve the fit of the model, it was decided to remove the relationship pathway between health problems and loneliness.

### 3.3. Structural Model of Mediated Relationships

The structural model of mediated relationships presented satisfactory goodness-of-fit indexes (WLSMV-χ^2^ (*df* = 345) = 875.106, *p* < 0.001; CFI = 0.992; TLI = 0.991; RMSEA = 0.034 [C.I. 90% = 0.031–0.037]). Figure 3 presents the model with the standardized coefficients, showing relationships involving direct and indirect influence on the dependent variable of loneliness. −

Figure 3 demonstrates direct, statistically significant negative relationships between health problems and resilience (β = −0.508; *p* < 0.001) and health problems and family functioning (β = −0.136; *p* < 0.001). Family functioning had a direct positive relationship with resilience (β = 0.388; *p* < 0.001).

Consistently with the above, there were direct and statistically significant negative relationships for resilience with loneliness (β = −0.321; *p* < 0.001) and with family functioning (β = −0.558; *p* < 0.001). As a result, it was the mediated relationships model that best fit the data (Figure 2).

## 4. Discussion

This study analyzed the psychosocial variables related to loneliness in old age for a mainly indigenous sample. A model was used with resilience and family functioning as mediating variables for health problems (Figure 1). The findings confirmed that loneliness among older adults is indirectly determined by chronic diagnosed pathologies and directly determined by their levels of resilience and the perception of a functional family environment (Figure 2 and Figure 3). These previous studies have confirmed that resilience makes it possible to age successfully by positively coping with the challenges and changes inherent to old age; it hence acts as a buffer against loneliness [46,47]. This is also observed in older adults who have a functional family environment that can properly meet their emotional and physical needs [48,49]. Health problems were indirectly related to loneliness in this study, confirming the particular features of successful ageing among ethnic minorities in Chile: as we will argue, health and psychosocial well-being in indigenous communities are the outcomes of balancing various dimensions. 

There was a high prevalence of loneliness in the multi-ethnic sample (Table 2) for a majority of the peoples analyzed, with prevalence equal to or greater than 55% except for Rapa Nui and Diaguita older adults (only 9% and 14%, respectively, of whom reported symptoms of loneliness). Pre-pandemic studies in Chile, which applied DJGLS-6, confirmed higher scores for the non-indigenous group (mean 3.99, standard deviation 1.71) than among the Aymara (mean 2.82, standard deviation 1.45) and Mapuche (mean 3.15, standard deviation 1.67) [36]. Meanwhile, other studies in non-indigenous populations report between 42% and 45% loneliness [33,35]. However, the national study by Herrera et al. [45] confirms an increase in loneliness during the pandemic among non-indigenous older people, from 48% (winter 2020) to 53% (autumn 2021).

This study supports the hypothesis of the impact of the COVID -19 pandemic on the loneliness of older adults [1,3,16]: their social relationships have been changed by the various social restrictions that were imposed on the general population, and with particular severity on older adults. The Chilean case also presents particular features in terms of the handling of the pandemic and its potential effects on well-being among older adults. Despite Chile having been one of the first countries in Latin America to vaccinate its population [50] and older adults having been prioritized in the vaccination process [51], Chile also maintained social restrictions for a longer period. In this regard, the Chilean Ministry of Health’s “Step by Step” (*Paso a Paso*) plan [52] took into account various epidemiological, healthcare, and care system capacity indicators for each region and/or municipality to change phase during the most difficult period of the pandemic. The phases ranged from full lockdown, consisting of strict confinement to the home except with a limited number of permits, to the “opening up” phase, in which it was permitted to leave the home during the week and there were more activities with established limits on capacity; these restrictions were more severe for older people. All of this meant that older adults experienced abrupt changes to their lifestyles, which affected their social and community relationships, causing a higher risk of social isolation [53].

Although loneliness in old age appears to be a transcultural problem [1,2], it can undoubtedly be determined by cultural factors that place greater or lesser emphasis on family or kin-based relationships [54]. In Chile, the family still plays a central role in the well-being of indigenous and non-indigenous older people [55,56]. This study can contribute to understanding the changes that Chilean indigenous communities are experiencing in their social, family, and reproductive organization, particularly in rural areas that host native settlements, as a result of migration by the younger or working population [57]. This has caused rapid depopulation and ageing in these territories [58]. In this regard, it has been observed in this study that many older people are living in single-member households and there is a trend involving a reduction in extended family structures where several generations cohabit (see Table 1). Changes to the system of family organization due to different household structures are endangering the traditional caregiving system [24], intergenerational transmission of knowledge, and indigenous traditions [25]. They are having a particular impact on emotional well-being, taking into account that the social exchanges that take place within the family are key to satisfying the emotional needs of indigenous older adults, protecting against loneliness, social isolation, and cognitive decline [26].

There has been extensive research into resilience in the context of old age [59,60], especially for high-risk groups such as indigenous communities [28,61], based on the premise that people are able to adapt and even emerge stronger after experiencing and coping with adverse, threatening, or highly stressful situations [62]. In this respect, resilience is not only a strategy for individual coping that is reinforced by intelligence, self-esteem, self-confidence, and self-efficacy [38], but it is also determined by the social and cultural context. In this respect, Lavretsky [63] maintains that cultural context determines how we construct our cosmovision, perception and understanding of ideas, morality, and preferences; as such, culture also impacts how we cope with trauma and adverse situations. Resilience has been investigated among indigenous peoples in Chile, despite historically facing an adverse socio-political context involving social inequality, they have been able to adapt to changes and reformulate their cultural reproduction and maintenance strategies [64]. In view of this, one may expect to observe that resilience mediates health problems and is related to fewer symptoms of loneliness.

As proposed in this study, family plays a central role in the well-being of older adults in Chile [33,56]. Family functioning is therefore also directly related to loneliness and mediates health problems. Family functionality is associated with the capacity of the family environment to provide a flexible, dynamic, and relevant response to physical needs, including reduced mobility, the risk of falls, and the need for specialist care, and emotional needs such as being heard, valued, respected, and loved [55,56]. Intimate and close social relationships have changed following the pandemic, negatively affecting emotional well-being in old age; this is why this study has produced such alarmingly negative data regarding the symptoms of loneliness.

Health problems were not directly related to loneliness, in contrast to the findings of previous studies [65,66]. This suggests a more holistic assessment of the concept of psychological well-being (loneliness) and health in the participating indigenous and Afro-descendant communities. In this regard, health is understood in indigenous communities as the harmony between the physical, mental, and spiritual aspects of a person. This harmony encompasses environmental, social, and cultural aspects [67]. Physical and mental health is a state of calm and balance in various dimensions. Therefore, when older adults suffer from chronic illnesses, they will activate their capacity for resilience and value the presence of a functional family environment that acts as a buffer against loneliness. This area specifically requires further investigation through qualitative and participative studies that involve indigenous communities and include a life-course approach to identify cumulative advantages and disadvantages over a lifetime.

The limitations of this study were notably related to the study methodology and the inclusion of an ethnic group only recently recognized by the Chilean State. In terms of methodological issues, the sample was not representative. Although valid inferences could be drawn thanks to quota sampling, representativeness is not guaranteed due to the lack of random selection. A further limitation was the lack of previous studies regarding the social realities faced by Chilean Afro-descendant older adults. The significance of this limitation may be low insofar as this study reports this social group’s experience of loneliness during the pandemic. 

## 5. Conclusions

Overall, loneliness is an issue that affects various vulnerable groups, but there is a greater prevalence where the intersectionality of ethnicity, rurality, and ageing intersects. In this study, we found that the psychosocial variables that are directly and indirectly related to loneliness are resilience, family functioning, and health problems. Older people’s resilience and family functioning can be intervened in earlier stages and even during old age; these are psychosocial variables that can be addressed in social policies. Gerontological interventions aimed at reducing loneliness should hence include strategies to reinforce the ability to cope and adjust (resilience) when confronted with stressful circumstances, as well as focusing on the social support that family can offer to older adults (family functionality).

## Figures and Tables

**Figure 1 ijerph-20-02138-f001:**
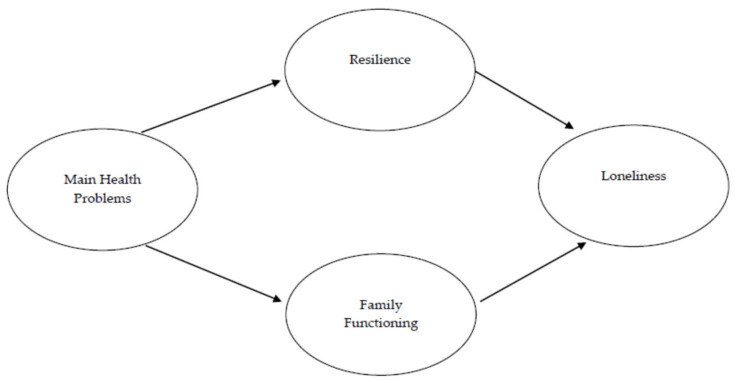
Hypothesized model.

**Figure 2 ijerph-20-02138-f002:**
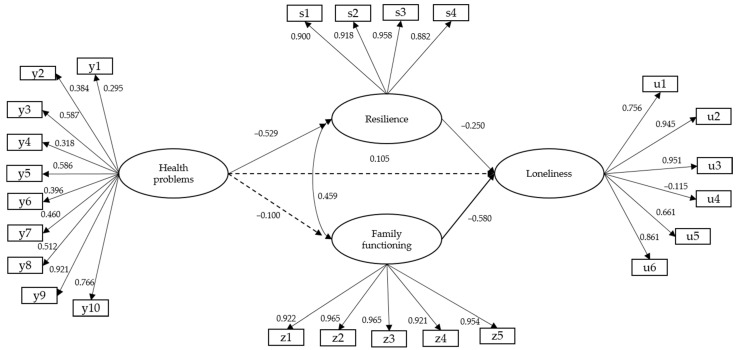
Complete model of structural relationships. Note: dotted line.

**Figure 3 ijerph-20-02138-f003:**
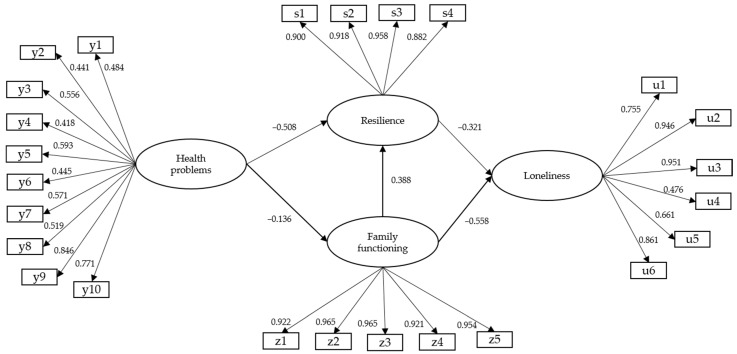
Model of structural relationships between health problems and loneliness, through resilience and family functioning.

**Table 1 ijerph-20-02138-t001:** Participant characteristics.

		Quechua (n = 203)	Atacameño (n = 214)	Colla (n = 215)	Chango (n = 100)	Diaguita (n = 255)	Rapa Nui (n = 130)	Huilliche (n = 116)	Kawésqar (n = 10)	Afro-Descendants (n = 105)	Total (1.348)
Gender	Women	123 (61)	152 (71)	96 (45)	53 (53)	145 (57)	71 (55)	67 (58)	8 (80)	59 (56)	774 (57)
	Men	80 (39)	62 (29)	119 (55)	47 (47)	110 (43)	59 (45)	49 (42)	2(20)	46 (44)	574 (43)
Age	60–69 years	105 (52)	109 (51)	85 (40)	48 (48)	104 (41)	91 (70)	43 (37)	6 (60)	67 (64)	658 (49)
	70–79 years	75 (37)	80 (37)	98 (46)	35 (35)	95 (37)	31 (24)	56 (48)	4 (40)	28 (27)	502 (37)
	80+ years	23 (11)	25 12)	32 (14)	17 (17)	56 (22)	8 (6)	17 (15)	0	10 (9)	188 (14)
Living situation	Alone	84(41)	52 (24)	52(24)	18(18)	79 (31)	25 (19)	32 (28)	1 (10)	11 (10)	69 (20)
	With someone	119 (59)	162 (76)	163 (76)	82 (82)	176 (69)	105 (81)	84 (72)	9 (90)	94 (90)	275 (80)
Marital status	Married/cohabiting	108 (53)	106 (50)	123 (57)	50 (50)	110 (43)	78 (60)	61 (61)	5 (50)	72 (69)	713 (53)
	Single	9 (4)	33 (15)	39 (18)	23 (23)	56 (22)	25 (19)	19 (16)	0	11 (10)	215 (16)
	Widow	67 (33)	53 (25)	39 (18)	13 (13)	78 (31)	27 (15)	21 (18)	5 (50)	15 (14)	318 (24)
	Divorced or similar	19 (10)	22 (10)	14 (7)	14 (14)	11 (4)	0	15 (13)	0	7 (7)	102 (7)
Education	Primary school incomplete	53 (26)	24 (11)	46 (21)	10 (10)	23 (9)	8 (6)	80 (69)	4 (40)	5 (5)	253 (19)
	Primary school	72 (35)	83 (39)	83 (39)	31 (31)	45 (18)	17 (13)	21 (18)	2 (20)	12 (11)	366 (27)
	High school or vocational education	76 (38)	100 (47)	74 (34)	53 (53)	169 (66)	88 (68)	15 (13)	4 (40)	71 (68)	650 (48)
	Higher education	2 (1)	7 (3)	12 (6)	6 (6)	18 (7)	17 (13)	0	0	17 (16)	79 (6)

Values are represented as n (%).

**Table 2 ijerph-20-02138-t002:** Prevalence of loneliness through DJGLS-6, by ethnic group.

	Afro-Descendants (n = 105)	Quechua (n = 203)	Atacameño (n = 214)	Colla (n = 215)	Chango (n = 100)	Diaguita (n = 255)	Rapa Nui (n = 130)	Huilliche (n = 116)	Kawésqar (n = 10)	Total (n = 1.348)
n(%)										
No loneliness (0–1 points)	s(45)	61(30)	54(25)	50(23)	38(38)	219(86)	118(91)	3(2)	2(20)	592(44)
Loneliness (≥2 points)	58(55)	142(70)	160(75)	165(77)	62(62)	36(14)	12(9)	113(98)	8(80)	756(56)

Values are represented as n(%).

**Table 3 ijerph-20-02138-t003:** Descriptive statistics for latent variables.

	Loneliness	Resilience	Health Problems	Family Functioning
Median	−0.260	0.286	−0.284	0.596
Minimum	−1.396	−4.043	−1.118	−3.205
Maximum	2.012	0.905	4.716	0.596
K-S Test	0.235 **	0.263 **	0.264 **	0.380 **

** *p* < 0.001, variables were standardized (average = 0, standard deviation = 1).

**Table 4 ijerph-20-02138-t004:** Pearson’s r correlation matrix.

	Loneliness	Resilience	Health Problems	Family Functioning
Loneliness	1			
Resilience	−0.421 **	1		
Health problems	0.138 **	−0.172 **	1	
Family functioning	−0.508 **	0.345 **	−0.065 **	1

** *p* < 0.001.

## Data Availability

Not applicable.

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
