# Peer review of "Loneliness and Psychosocial Resources among Indigenous and Afro-Descendant Older People in Rural Areas of Chile"

_ijerph, 2023, doi:10.3390/ijerph20032138_

Round 1
Reviewer 1 Report (Previous Reviewer 1)
The revised version has incorporated the comments of the reviewers and has corrected the main inconsistency detected.
Author Response
Thank you very much for your comments, we have added the small suggestions made in the section (This study) and we have tried to be clearer about the age criteria to consider "older person" in Chile.
Reviewer 2 Report (Previous Reviewer 2)
Dear Authors,
The manuscript was improved.
The "this study section" is an English edition question! I suggest a native English edition to overcome the unclear meaning. The authors should clarify that older adults correspond to an age range of upper to 60 years old.
Author Response
Thank you very much for your comments, we have added the small suggestions made in the section (This study) and we have tried to be clearer about the age criteria to consider "older person" in Chile.
This manuscript is a resubmission of an earlier submission. The following is a list of the peer review reports and author responses from that submission.
Round 1
Reviewer 1 Report
This research aims to analyze the psychosocial variables related to loneliness in old age in a multi-ethnic sample in Chile (n=1.692). Structural equation modeling (SEM) showed indirect relationships between health problems via family functioning, resilience, and loneliness.
General comment: On the one hand, the sample is multi-ethnic, and the data analysis is carried out on this complete sample, which, although primarily indigenous peoples, includes around a fifth of non-indigenous people. On the other hand, the justification, the reading of the results and the discussion emphasize the indigenous. Therefore, there is a contradiction here. For greater consistency in the article, the SEM models should be redone, including only the sample of native peoples or comparing these models with that of non-indigenous people.
Title: if the sample includes older people in rural settings, it should be stated in the title that it is about this.
Methodology:
More information is required on the sample selection method. For example, representativeness is spoken of in a broad sense but not statistically. How were the people selected? Was there some method of random and probabilistic selection? Was the sample selected intentionally? Where were the people interviewed?
It is also unclear whether the application was entirely face-to-face or who administered the questionnaires (this information is indicated only for the pretest).
Regarding the sample, it is striking that, being a rural sample, the educational level does not seem to be below the general level of the Chilean older adult population. Some comments should be made regarding the validity of the composition of this sample, comparing it with some population parameters, for example, the last National Census of Chile.
It does not make sense to calculate Cronbach's Alpha for the variable "main health problems", since it is an index that adds health problems. The presence of a problem does not have to be correlated with the presence of other health problems.
Discussion: when commenting on how the Covid-19 pandemic was experienced in Chile, it should be explicitly noted that the elderly had forced confinement in some periods only for this age group, in a context where restrictions began to be relaxed for the other age groups.
Lines 372-374. The following cannot be pointed out because the results are presented for the complete sample and not only for the indigenous people. In addition, since it is not a longitudinal study, changes over time cannot be evaluated: “As we have seen in this study, indigenous communities in Chile are experiencing significant changes to their social, family and reproductive organization, particularly in rural areas…”.
You have to review the conclusions. The conclusion of the paper states that “this study confirms the existence of cultural differences in the experience of loneliness in old age among Chilean ethnic minorities, as well as the presence of psychosocial variables that could be subject to intervention to promote emotional wellbeing. This study also shows the negative impact of the Covid-19 pandemic on loneliness.
The first of these statements cannot be inferred from the results of the study, since the levels of loneliness or its predictors are not compared between different ethnic minorities in the sample. The last conclusion is incorrect either because the study does not compare it with the experience of loneliness before the Covid-19 pandemic. As it is not a longitudinal study, it cannot be stated that the pandemic had a negative impact on loneliness.
Please review the following wordings:115-118
159-161
201-206
337 resilience
Data source missing in lines 153-156.
Author Response
The authors are grateful for the suggestions made, which will undoubtedly lead to a substantial improvement of the paper. Please note that the changes made to the text and bibliography are in red.
REVIEWER1
This research aims to analyze the psychosocial variables related to loneliness in old age in a multi-ethnic sample in Chile (n=1.692). Structural equation modeling (SEM) showed indirect relationships between health problems via family functioning, resilience, and loneliness.
(1) General comment: On the one hand, the sample is multi-ethnic, and the data analysis is carried out on this complete sample, which, although primarily indigenous peoples, includes around a fifth of non-indigenous people. On the other hand, the justification, the reading of the results and the discussion emphasize the indigenous. Therefore, there is a contradiction here. For greater consistency in the article, the SEM models should be redone, including only the sample of native peoples or comparing these models with that of non-indigenous people.
- Taking into account that the aim of this paper is to analyse the psychosocial variables related to loneliness in old age in a multi-ethnic sample in Chile (n=1.692). It does not seek to make a specific differentiation by ethnic groups and it is understood that the data are mostly from indigenous population. In addition, it should be considered that structural equation models (SEM) are multivariate models that require a large sample size, at least 10 individuals for each estimated item (Nunnally, 1967), this requirement makes it unlikely to be able to estimate a model for each item ethnic condition (10 different conditions) and expect stability in the parameters. However, following the recommendations of the reviewer we have included background information on loneliness in non-indigenous older people in Chile, however, these data are from people living in urban areas.
- In this way, the authors believe that the value of this work is the particularity of the sample and its analysis as a whole. Therefore, we will maintain the model proposed for the total multi-ethnic sample. To avoid possible inconsistency, we have included the available data on loneliness (pre-pandemic and post-pandemic) in non-indigenous populations.
(2) Title: if the sample includes older people in rural settings, it should be stated in the title that it is about this.
- DONE
(3) Methodology:
More information is required on the sample selection method. For example, representativeness is spoken of in a broad sense but not statistically. How were the people selected? Was there some method of random and probabilistic selection? Was the sample selected intentionally? Where were the people interviewed?
It is also unclear whether the application was entirely face-to-face or who administered the questionnaires (this information is indicated only for the pretest).
Regarding the sample, it is striking that, being a rural sample, the educational level does not seem to be below the general level of the Chilean older adult population. Some comments should be made regarding the validity of the composition of this sample, comparing it with some population parameters, for example, the last National Census of Chile.
- Thank you very much for this comment, we have incorporated more information about the sample and the application process. We have established the various difficulties in gaining access to interview more than a thousand older people in rural areas in a period of pandemic restrictions, so we do not believe that the sample loses validity due to the level of study of the subjects interviewed, it is a sample of convenience and voluntary participation. We believe that by incorporating more background on the sampling procedure, the value of its face-to-face application and by qualified professionals in areas of complex access (as well as the various limitations overcome by the research team), we provide clarity on the rationale for the value of non-probability sampling.
It does not make sense to calculate Cronbach's Alpha for the variable "main health problems", since it is an index that adds health problems. The presence of a problem does not have to be correlated with the presence of other health problems.
- DONE
(3) Discussion: when commenting on how the Covid-19 pandemic was experienced in Chile, it should be explicitly noted that the elderly had forced confinement in some periods only for this age group, in a context where restrictions began to be relaxed for the other age groups.
- DONE
(5) Lines 372-374. The following cannot be pointed out because the results are presented for the complete sample and not only for the indigenous people. In addition, since it is not a longitudinal study, changes over time cannot be evaluated: “As we have seen in this study, indigenous communities in Chile are experiencing significant changes to their social, family and reproductive organization, particularly in rural areas…”.
- The authors are aware of the limitations of the study, which is why we have indicated at the end of the discussion that we cannot establish causal relationships, but we can observe significant relationships between the variables analysed.
- With regard to the discussion focused on indigenous older people, we indicate that loneliness among older people is cross-cultural. We also add background information on loneliness among non-indigenous people. In this sense, we begin the paragraph by indicating that loneliness affects cross-culturally and the data of this research show a change in the structure of the household that negatively influences the system of social, family and reproductive organisation of indigenous people, all of which is supported by empirical evidence at the international level.
(6) You have to review the conclusions. The conclusion of the paper states that “this study confirms the existence of cultural differences in the experience of loneliness in old age among Chilean ethnic minorities, as well as the presence of psychosocial variables that could be subject to intervention to promote emotional wellbeing. This study also shows the negative impact of the Covid-19 pandemic on loneliness.
The first of these statements cannot be inferred from the results of the study, since the levels of loneliness or its predictors are not compared between different ethnic minorities in the sample. The last conclusion is incorrect either because the study does not compare it with the experience of loneliness before the Covid-19 pandemic. As it is not a longitudinal study, it cannot be stated that the pandemic had a negative impact on loneliness.
- DONE
(6) Please review the following wordings:115-118; 159-161; 201-206; 337 resilience
- DONE
(7) Data source missing in lines 153-156.
- DONE
Reviewer 2 Report
Dear Authors,
In general, the manuscript focuses on an exciting topic!
Some suggestions should be pointed out:
The author should be identified in the text related to Lines 39 to 41 (ref 8) and in lines 50-51. The designation "older adults" should be clarified! (line 166) Line 187, "average age of 70.49 years, standard deviation=7.39," should be replaced with 70,49 ± 7,39. Table legends should include the sample size. Did the authors intend to report % value?
Author Response
The authors are grateful for the suggestions made, which will undoubtedly lead to a substantial improvement of the paper. Please note that the changes made to the text and bibliography are in red.
REVIEWER 2
In general, the manuscript focuses on an exciting topic!
Some suggestions should be pointed out:
The author should be identified in the text related to Lines 39 to 41 (ref 8) and in lines 50-51. The designation "older adults" should be clarified! (line 166) Line 187, "average age of 70.49 years, standard deviation=7.39," should be replaced with 70,49 ± 7,39. Table legends should include the sample size. Did the authors intend to report % value?
- DONE. We have incorporated all the suggestions. It should be noted that in Chile, as in the rest of Latin America, the study of the elderly begins at the age of 60. This is explicitly stated in the section "this study".
Reviewer 3 Report
Resultados de tradução
Abstract Mention the name of the scales used in the method the type of study In the result, present the value of the statistical analysis In the conclusion, it makes several statements and some are not considered as objectives of the study, review the objective or conclusion Introduction: It problematizes the indigenous people, it must do the same with the other analyzed groups Method: Mention the various geographical regions analyzed, was a sample calculation made for each region or was it for convenience Quote the deletion protocol Table 1 should appear in results The application was face-to-face or remote Make the Cronbach's index for all instruments used As mentioned in the others, cite the Family APGAR validation for chile Results I suggest carrying out a comparative analysis between the analyzed tool groups, as well as regression analysis in order to show a possible predictive association between the variables and the instruments Discussion Include discussion of the results measured by the suggestion of adding analysis Conclusion: Same conclusion of the abstractAuthor Response
Extensive editing of English language and style required: We would like to point out to the reviewer that this work has been translated and revised by a native English speaker (Steve Churnin) with whom we have been working for twelve years and with whom we have had no difficulties in the process of editing into English, not only because he is a native speaker, but also because he is a professional in the psychosocial area and in the fields of study: https://www.proz.com/profile/1654094
The authors are grateful for the suggestions made, which will undoubtedly lead to a substantial improvement of the paper. Please note that the changes made to the text and bibliography are in red.
(1) Abstract Mention the name of the scales used in the method the type of study In the result, present the value of the statistical analysis. In the conclusion, it makes several statements and some are not considered as objectives of the study, review the objective or conclusion
- DONE
(2) Introduction: It problematizes the indigenous people, it must do the same with the other analyzed groups.
- This comment is very pertinent as the sample is multi-ethnic, we have briefly included background information on loneliness in the older non-indigenous population in the introduction and discussion. It should be noted that the value of this study lies in the fact that 80% of the sample is indigenous, which is why the work deals in more depth with ageing and the problem of loneliness in this social group, which has been scarcely researched in Chile.
(3) Method: Mention the various geographical regions analyzed, was a sample calculation made for each region or was it for convenience Quote the deletion protocol Table 1 should appear in results The application was face-to-face or remote Make the Cronbach's index for all instruments used As mentioned in the others, cite the Family APGAR validation for Chile.
- We have detailed the sampling and implementation process. The sample was selected by quotas (by convenience and voluntariness of the elderly) of the ethnic groups and the regions studied are justified because they are "the native places of each ethnic group". The study was conducted face-to-face, now indicated in the methodology.
- We believe that Table 1 is best illustrated in the description of the sample, so we will keep it in the methodology section.
- Following the indications of reviewer 1 (It does not make sense to calculate Cronbach's Alpha for the variable "main health problems", since it is an index that adds health problems. The presence of a problem does not have to be correlated with the presence of other health problems) we have eliminated the alpha for health problems.
- The validation of the APGAR in Chilean elderly people is incorporated.
(4) Results I suggest carrying out a comparative analysis between the analyzed tool groups, as well as regression analysis in order to show a possible predictive association between the variables and the instruments.
- We appreciate the reviewer's comment, structural equation models (SEM) are multivariate models that come from the family of regression models, however, mathematically they are more complex, flexible, and appropriate for the variables studied (categorical latent variables). SEM models are born to provide regression models with more flexibility in mathematical assumptions, also adding to the model the possibility of estimating measurement errors in both dependent and independent variables. In addition, the model represented in Figure 2 presents a dependent variable (Loneliness) and the direct effects of the independent variables (Health problems, family functioning and resilience), this model being equivalent to a regression model that estimates all direct effects with the dependent variable. Table 4 shows the relationships between the study variables.
(5) Discussion Include discussion of the results measured by the suggestion of adding analysis
- We have incorporated some background information on the non-indigenous Chilean population, but as we have stated, our interest is to show with this study how loneliness is affecting vulnerable groups that have been little addressed in studies in Chile: indigenous ethnic minorities and Afro-descendants.
(6)Conclusion: Same conclusion of the abstract
- We have modified the conclusion in the abstract (more direct) and also in the manuscript. We address each of the variables analyzed in the model: resilience, family functioning and health problems.